# Headache Because of Problems with Teeth, Mouth, Jaws, or Dentures in Chronic Temporomandibular Disorder Patients: A Case–Control Study

**DOI:** 10.3390/ijerph19053052

**Published:** 2022-03-05

**Authors:** Tadej Ostrc, Sabina Frankovič, Zvezdan Pirtošek, Ksenija Rener-Sitar

**Affiliations:** 1Department of Dental Prosthetics, Dental Division, Faculty of Medicine, University of Ljubljana, 1000 Ljubljana, Slovenia; ksenija.rener@mf.uni-lj.si; 2Department of Prosthetic Dentistry, Division of Stomatology, University Medical Center Ljubljana, 1000 Ljubljana, Slovenia; 3Mental Health Dispensary, Community Health Centre Kranj, 4000 Kranj, Slovenia; sabina.frankovic@gmail.com; 4Department of Neurology, University Medical Center Ljubljana, 1000 Ljubljana, Slovenia; zvezdan.pirtosek@kclj.si; 5Department of Neurology, Faculty of Medicine, University of Ljubljana, 1000 Ljubljana, Slovenia

**Keywords:** temporomandibular disorders, headache, depression, anxiety, somatization, sleep quality

## Abstract

This study aimed to characterize self-reported headaches because of problems with the teeth, mouth, jaws, or dentures (HATMJD) in chronic patients with temporomandibular disorders (TMDs) in order to compare their results with those of TMD patients without such headaches and to investigate the associations of HATMJD with depression, anxiety, physical symptoms, oral behaviors, and sleep quality. We conducted a case–control study on consecutive chronic TMD patients referred to the University Medical Center of Ljubljana, Slovenia. A self-reported HATMJD was extracted from item #12 in the 49-item version of the Oral Health Impact Profile questionnaire. Axis II instruments of the Diagnostic Criteria for TMD (i.e., for screening of depression, anxiety, specific comorbid functional disorders, and oral behaviors) and the Pittsburgh Sleep Quality Index were used in this study. In total, 177 TMD patients (77.4% women; mean age: 36.3 years) participated in this study; 109 (61.6%) patients were classified as TMD patients with HATMJD. TMD patients with at least mild depressive and anxiety symptoms, with at least low somatic symptom severity, and a high number of parafunctional behaviors had more HATMJD. Parafunctional behavior and sleep quality were the most prominent predictive factors of the occurrence of HATMJD. TMD patients with HATMJD have more psychosocial dysfunction, a higher frequency of oral behaviors, and poorer sleep quality than TMD patients without such headaches.

## 1. Introduction

Many clinical studies have confirmed associations between temporomandibular disorders (TMD) and tension-type headaches or migraine [1,2,3,4]. TMD and headaches may have common pathogenesis, causation, or common disruptive factors [3,5,6]. Furthermore, musculoskeletal pain in the orofacial area is more common in people with headaches [7]. In addition, TMD in patients with severe headaches have more significant pain intensity [7,8,9,10,11,12].

It is generally accepted that TMDs have a multifactorial etiology and that psychosocial components, including oral behaviors and trauma, contribute to the development, exacerbation, and progression to chronic TMD [13,14]. However, the role of psychological factors in the development of TMD disorders is still not entirely clear [15]. Some studies have concluded that patients with chewing muscle pain are more prone to stress and depression [16,17]. Stress and anxiety are the well-known psychological factors that correlate with TMD because stress can cause muscle hyperactivity, resulting in symptoms of TMD [15,18].

Psychological distress (e.g., anxiety, depression, and specific comorbid functional disorders) increases the symptom burden and functional impairment in several chronic medical conditions (e.g., headaches) [19,20,21,22]. For example, approximately half of acute TMD patients and 10% of headache sufferers have comorbid anxiety disorders, indicating that anxiety may be an early feature in both patient populations [20].

Psychopathology may also discriminate between myogenous and arthrogenous groups of TMD patients [14]. According to Gatchel et al., depression is more common among individuals with chronic tension-type headaches than with TMD [21]. Furthermore, the study by van der Meer et al. has shown that the associations between self-reported headache and painful/function-related TMD were confounded by the presence of somatic symptom complaints [22].

Oral parafunctions include sleep and awake bruxism, lip biting, thumb sucking, and any other oral habit not associated with mastication, deglutition, and speech [23]. The most common oral parafunctional activity is sleep bruxism, with a prevalence of up to 90% in the general population [24]. The relationship between oral parafunctions and TMD signs and symptoms has been reported in more studies [25,26]. An association between headache and bruxism has also been extensively explored [27]. Glaros et al. assessed oral parafunctions in headache and non-headache control groups and reported that headache patients had significantly more frequent and intense tooth contacts, more masticatory muscle tension, and more stress than non-pain controls [28]. Another study by Glaros et al. showed that subjects with headaches were significantly more likely to receive a diagnosis of masticatory myofascial pain than those in the non-headache control [29].

The substantial relationship of myofascial pain on sleep quality is documented in the study, which assessed differences in sleep quality between patients with chronic daily headaches and patients with TMD [30]. It was found that sleep quality was significantly worse in TMD patients with myofascial pain than patients with chronic daily headaches and intracapsular pain [30]. Therefore, assessing sleep quality must also be considered for TMD patients who are chronically distressed by their condition and is highly recommended in patients with dysfunctional TMD pain [31]. Sleep disturbances and headache disorders have similar pathogenic mechanisms; therefore, tension-type headaches, migraines, and sleep disturbances often occur together [32,33]. Insomnia is a known risk factor for headaches, especially tension-type headaches and migraines [34]. Sleep disruption is a common problem among subjects reporting headaches, and it is reported that sleep quality exhibits a complex interaction in individuals with chronic tension-type headaches [35]. Other studies have reported that poor sleep quality is not correlated with migraine [36]. A recent study has shown that patients with more severe insomnia present more severe depressive and anxiety symptoms [37].

The diagnosis of “headache attributed to temporomandibular disorder” (HATMD) was first introduced as a secondary headache in the second edition of International Classification of Headache Disorders (ICHD-II) under the code 11.7 by the Headache Classification Subcommittee of the International Headache in 2004 and further defined in the third version (i.e., ICHD-III) in 2018 [38]. The HATMD diagnosis is defined as headache and facial pain due to problems in the temporomandibular joints (TMJs), masticatory muscles, and/or associated structures after all primary headaches are excluded [38]. Similarly, the “Diagnostic Criteria for Temporomandibular Disorders” (DC/TMD) also include HATMD, which further implies that orofacial musculoskeletal pain and/or TMD are associated with headache [39]. The temporal region, preauricular area, and/or masseter muscle are commonly tender in HATMD [39]. Unilateral or bilateral HATMD follows the pattern of affected ipsilateral or bilateral temporomandibular regions, respectively [38].

Primary headaches (e.g., migraine and tension-type headaches) contribute to TMD problems [27]. Conversely, muscle-related TMD are highly associated with the presence of migraine [40,41]. Although many studies explored associations between psychological distress, oral behaviors and/or sleep quality and TMD, or the same factors in patients with HATMD, none have explored the extent of associations of self-reported headache because of problems with teeth, mouth, jaws, or dentures (HATMJD), with psychological distress, oral behaviors, and sleep quality in TMD patients, which is a clinically relevant question. Therefore, this study aimed to characterize self-reported HATMJD in chronic TMD patients, to compare their results with those of TMD patients without such headaches, and investigate the associations of HATMJD with depression, anxiety, physical symptoms, oral behaviors, and sleep quality.

## 2. Materials and Methods

### 2.1. Ethical Approval

This research was approved by the Faculty of Arts’ Ethics Committee, University of Ljubljana, Slovenia, on 6 June 2016, and by the National Medical Ethics Committee of the Republic of Slovenia (KME 124/05/16). All participants signed a statement regarding informed consent before beginning the study.

### 2.2. Observed Population

This case–control study was performed on consecutive adult chronic TMD patients referred for TMD management to the Clinic for Orofacial Pain and Dental Sleep Medicine within the Department for Prosthetic Dentistry (University Dental Clinics, University Medical Center Ljubljana, Slovenia) between March 2016 and February 2021. Exclusion criteria included being aged less than 18, the presence of orofacial pain disorders other than TMD, or systemic rheumatic diseases. The diagnosis of TMD was based on history and physical examination findings according to DC/TMD protocol by a board-certified specialist for orofacial pain. The baseline data prior to treatment were analyzed for this study. 

### 2.3. Study Instruments

We collected the data about demographics, such as gender, age, marital status, and level of education. The presence of the self-reported HATMJD was extracted from the Slovenian version of the psychometrically validated 49-item Oral Health Impact Profile questionnaire (OHIP), specifically from item #12 (“Have you had headaches because of problems with your teeth, mouth, jaws or dentures?”) [42]. Responses regarding the HATMJD were rated on the following scale: 0 = “Never”, 1 = “Hardly ever”, 2 = “Occasionally”, 3 = “Fairly often”, and 4 = “Very often” [43].

Additionally, Axis II instruments of the Diagnostic Criteria for TMD (i.e., for screening of depression, anxiety, specific comorbid functional disorders, and oral behaviors) and the Pittsburgh Sleep Quality Index were used in this study. All these questionnaires were translated into the Slovenian language and psychometrically validated in the Slovenian TMD population in the previous study [44].

*Depression.* PHQ-9 was introduced in 2001 and is designed to screen and measure the severity of depression [44]. Patients are asked how often they were bothered by problems over the last two weeks. Answers are formulated on a scale from 0 (not at all) to 3 (nearly every day). The total score for the nine items ranges from 0 to 27. Higher scores represent a higher level of depression. Scores of 5, 10, 15, and 20 represent the cut-off points for mild, moderate, moderately severe, and severe depressive symptoms, respectively. Cronbach’s alpha for measurement of internal consistency for the Slovenian version was 0.75 when assessed in Slovenian TMD patients [45].

*Anxiety.* GAD-7 was introduced in 2006 as a self-reported questionnaire to screen and measure generalized anxiety disorder severity [46]. Patients are asked how often they were bothered by problems over the last two weeks. Answers are formulated on a scale from 0 (not at all) to 3 (nearly every day). The total score for the seven items ranges from 0 to 21. A higher score represents a higher level of anxiety. Scores of 5, 10, and 15 are the cut-off points for mild, moderate, and severe anxiety, respectively. The internal consistency of this questionnaire for the Slovenian version is high when assessed in Slovenian TMD patients, with a Cronbach’s alpha of 0.88 [45].

*Physical symptoms.* PHQ-15 was introduced in 2002 to inquire about 15 somatic symptoms (i.e., specific comorbid functional disorders) that account for more than 90% of the physical complaints [47]. Subjects are asked to rate the severity of each symptom as 0 (not bothered at all), 1 (bothered a little), or 2 (bothered a lot) over the previous week. The summary score ranges from 0 to 30. A higher score indicates a higher level of somatic symptom severity. Scores of 5, 10, and 15 are the cut-off points for low, medium, and high somatic symptom severity, respectively. The Cronbach’s alpha for the Slovenian version is 0.69 when assessed in Slovenian TMD patients [45].

*Parafunctions.* The Oral Behavior Checklist (OBC) is a self-report scale for identifying and quantifying the frequency of jaw overuse behaviors (e.g., grinding, clenching teeth together, tightening and tensing the masticatory muscles, abnormal jaw posture, intense pressing of the tongue against the teeth, excessive talking, yawning) [48]. For each item, the subject is asked to report the frequency of the occurrence of specific oral behaviors over the past month, using the response options “none of the time”, “a little of the time”, “some of the time”, “most of the time”, and “all of the time” on a scale from 0 to 4. The summary score ranges from 0 to 84. A higher score indicates a higher number of parafunctional activities. A score of 0 indicates no parafunctional activity. Scores up to 24 indicate a low frequency of jaw overuse behaviors, whereas scores of 25 or more indicate a high frequency of jaw overuse behaviors.

*Sleep quality.* The Pittsburgh Sleep Quality Index (PSQI) measures the self-reported quality and patterns of sleep in adults [31]. It differentiates “poor” from “good” sleep by measuring seven areas: subjective sleep quality, sleep latency, sleep duration, habitual sleep efficiency, sleep disturbances, use of sleeping medication, and daytime dysfunction over the last month. The patient self-rates these seven areas of sleep. The answers are based on a 0 to 3 scale, in which 3 reflects the negative extreme on the Likert scale. Summary scores range from 0 to 21, and scores higher than 5 indicate poor sleep quality. The Slovenian version of PSQI has a Cronbach’s alpha reliability coefficient of 0.74 for its seven components [49].

### 2.4. Data Collection

All questionnaires were administered to our TMD patient cohort in electronic form before treatment. The questionnaires were presented in 1KA software [50], a free, open-source application for online surveys. In addition, each patient received a unique and anonymous code to register on the platform to complete the questionnaires at home.

For this study purpose, the TMD patients were subsequently divided according to the presence or absence of the self-reported HATMJD, based on item #12 from the OHIP questionnaire results. If the TMD patients responded to item #12 with “Never” or “Hardly ever”, we interpreted this as the absence of HATMJD. In contrast, if they answered to this item as “Occasionally”, “Fairly often”, or “Very often”, we interpreted this as the presence of HATMJD. Therefore, the TMD cohort was stratified into the HATMJD and non-HATMJD groups, which were our study and control groups, respectively.

### 2.5. Data Analysis

The variables analyzed as potential explanatory factors responsible for the variability of our observed outcome (i.e., the presence of HATMJD) were patient demographics, depression, anxiety, specific comorbid functional disorders, parafunctional behaviors, and sleep quality. Because measures for our constructs were continuous or ordinal in their original metric (e.g., the Likert-type scale), our TMD patients were grouped according to DC/TMD recommendations as follows: depression (no, mild, moderate, moderately severe, or severe depressive symptoms), anxiety (no, mild, moderate and severe anxiety), specific comorbid functional disorders (no, low, medium or high somatic symptom severity), and parafunctional behaviors (none, low or high) [51]. Sleep quality was grouped as good or poor (i.e., a PSQI score greater than five was considered poor sleep quality). According to Colarusso, the young adult population was defined as below 40 years of age [52].

The effect sizes with 95% confidence intervals (i.e., Cohen’s d) demonstrated the standardized differences between the two groups. Standard guidelines for interpreting effect size suggest that 0.2 is considered small, 0.5 medium, and 0.8 a large effect [53]. The associations between our observed outcome (i.e., the presence of HATMJD, and explanatory factors, e.g., depression, anxiety, specific comorbid functional disorders, parafunctional behaviors, and sleep quality) were first assessed with simple logistic regression analysis. Subsequently, the explanatory factors, which proved statistically significant, were entered in multiple logistic regression analysis, with stepwise forward variable selection (likelihood ratio) with HATMJD presence as the outcome to identify the significant associations between the predictors, which were independent variables and our observed outcome, being dependent variable, which is the presence of HATMJD.

The model’s predictive power was evaluated using the Omnibus Test of Model, and the Hosmer–Lemeshow test was used to examine the goodness of the model’s fit [54]. Nagelkerke’s R-square was generated to express the proportion of variance explained by the model [55]. The odds ratio (OR) and 95% confidence intervals (95% CI) were calculated in simple logistic regression and multiple logistic regression models. Statistical significance was set at the level of *p* < 0.05. Statistical analysis was performed in SPSS for Windows software, version 25.0 (IBM Corporation, Armonk, NY, USA).

## 3. Results

One hundred and seventy-seven adult TMD patients (137 women; patient cohort mean age: 36.3 years) participated in this study. All participants answered all items in the previously described questionnaires. Of all included TMD patients, 109 (61.6%) patients were classified as TMD patients with HATMJD, whereas 68 (38.4%) TMD patients reported no presence of HATMJD. The demographic and clinical characteristics of the studied TMD patient cohort and its subgroups (i.e., the study group of TMD patients with a HATMJD) and the second subgroup (i.e., the control group of TMD patients without self-reported HATMJD) are presented in Table 1.

Table 2 displays the analysis of the effect sizes (i.e., influences of sociodemographic variables and selected DC/TMD Axis II measures on HATMJD) using standardized mean effect reported as Cohen’s d.

The results of the simple logistic regression analysis are presented in Table 3. 

The complete five-step multiple logistic regression analysis is shown in Table 4. 

The final model had a statistically significant predictive power (χ^2^ = 32.399, *p* < 0.001; Hosmer–Lemeshow test for goodness of fit: χ^2^ = 10.206, *p* = 0.251), and overall, correctly classified 70.6% of the TMD patients. The final model explained 22.7% of the variance for the presence of HATMJD (Nagelkerke’s R square was 0.227). 

## 4. Discussion

Our study showed that TMD patients with a self-reported HATMJD are mainly women, with greater chance of psychological dysfunction, higher frequency of oral behaviors, and poorer sleep quality than TMD patients without such headaches. Statistically significant factors from simple logistic regression analysis (i.e., gender, depression, anxiety, specific comorbid functional disorders, oral behaviors, and sleep quality) were included in multiple logistic regression analyses to identify the significant associations between the predictors (independent variables) and the outcome (dependent variable: the presence of a HATMJD). In other words, we aimed to create the best regression model from our predictor variables that most accurately explain the associations between our predictor variables and the presence of HATMJD.

Our TMD patients with a high frequency of parafunctional activities had a 1.2-times higher chance of developing the self-reported HATMJD. Furthermore, in female TMD patients, the perceived HATMJD was 2.6-times more likely to occur than in male TMD patients. The female gender was the most prominent statistically significant sociodemographic characteristic in TMD patients with HATMJD. These findings are consistent with previous studies that investigated the co-morbidity of TMD and headaches [7,22,56,57]. Therefore, the self-reported HATMJD rate was statistically more frequent in female TMD patients than in male ones, and, when expressed as the effect size, it was medium. There was no statistically significant difference between the HATMJD group versus the non-HATMJD group for marital status and education. Additionally, the high number of parafunctional behaviors correlated statistically significantly with the presence of HATMJD, with an effect size close to large. The HATMJD presence was statistically higher in people with poorer sleep quality than in people with good sleep quality, and when expressed as effect size, it was medium.

Medium to high somatic symptom severity was reported in 34%, moderate to severe anxiety in 19.8%, and moderate to severe depression in 19.3% of TMD patients, who reported HATMJD in our study. The association between psychological distress (including depression, specific comorbid functional disorders, and anxiety) and TMD pain was investigated in some previous studies [58,59]. In the systematic review from De La Torre Canales et al., in which they included 14 studies investigating psychosocial impairment in TMD patients, the prevalence of medium to high somatic symptom severity varied from 28.5% to 76.6% and for moderate to severe depression from 21.4% to 60.1% [60]. It is important to emphasize that different study instruments for psychological distress were used in our study (i.e., PHQ-9 and PHQ-15) than in the study of De La Torre Canales et al., in which they utilized the instruments SCL-90-SOM and SCL-90-DEP, which could explain the differences in prevalence.

In the final model adjusted for age and gender, three variables were statistically significantly correlated with the presence of HATMJD in our TMD cohort. Female patients with poorer sleep quality and a higher frequency of parafunctional oral behaviors were more likely to have the presence of HATMJD. Therefore, oral behaviors and sleep quality when adjusted for gender are the statistically significant factors contributing to the perceived HATMJD. Our findings, therefore, are consistent with previous studies in patients with tension-type headaches and migraines [29,32], which showed that headache patients are significantly more likely to report oral parafunctional behaviors and that poor sleep quality is associated with higher severity of headaches. In addition, it is known that parafunctional oral behavior could effectively be addressed with early biobehavioral intervention [61,62]. Such treatment could also potentially alleviate the self-reported headache in the TMD patient population [29].

Interestingly, in the recent study by Reiter et al., which retrospectively evaluated DC/TMD Axis I and Axis II data from 220 patients having pain-related TMD with HATMD, the mean depression scores measured with PHQ-9 were similar (6.88 ± 4.69) to our study (6.6 ± 5.3), mean anxiety scores, i.e., GAD-7 scores, were lower (4.35 ± 3.79) than in our study (4.9 ± 4.8), and somatic symptom severity, i.e., PHQ-15 scores, were also lower (7.18 ± 4.18) than in our study (8.4 ± 4.4) [63]. Although we included all TMD patients in our study and not only those with pain-related TMD, we still had higher anxiety and somatic symptom severity scores in comparison to the study by Reiter et al. [63], who included only patients with pain-related TMD and with comorbid HATMD. 

The study of Tchivileva et al. reported a mean PSQI summary score of 6.0 in patients with comorbid tension-type headache and HATMD and 7.6 in patients with comorbid migraine and HATMD [64]. In contrast, in our study, the average sleep quality, reported as the PSQI score, was 6.8 for our TMD patients with HATMJD. Therefore, if we compare our patients’ sleep quality with the sleep quality results by Tchivileva et al. [64], we see that our sleep quality results were exactly in between their sleep quality results for TMD patients with tension-type headaches and TMD patients with migraine. This finding is not surprising because HATMJD probably included at least some primary headaches.

To our knowledge, this is the first study to investigate the associations between all types of self-reported HATMJD in the TMD patient cohort and the psychosocial dysfunction, sleep quality, and parafunctional activities. Nevertheless, this study has limitations. We did not specify which TMD diagnosis is more associated with self-reported HATMJD. Our study aimed to focus on psychosocial predictors for HATMJD in TMD patients, and for this reason, we did the multiple logistic regression analysis on the complete TMD patient cohort. Additionally, the accuracy of self-perceived oral parafunctional behaviors is questionable when reported only using a questionnaire because most individuals with sleep bruxism are not aware of this oral behavior since it occurs during sleep.

## 5. Conclusions

TMD patients with at least mild depressive and anxiety symptoms, with at least low somatic symptom severity, and a high number of parafunctional behaviors, had more HATMJD. This is additional proof that the DC/TMD Axis II constructs are crucial outcome predictors for TMD treatment purposes and HATMJD in this patient population. Furthermore, this study indicates that besides HATMD, which is a secondary headache, other types of headaches, including the primary ones, are also associated with the DC/TMD Axis II constructs and sleep quality when present in TMD patients. Therefore, assessing psychological distress, oral behaviors, and sleep quality is highly recommended in TMD patients with comorbid headache(s). Future research could also focus on how and if various TMD treatment modalities also affect different types of headaches in TMD patients.

## Figures and Tables

**Table 1 ijerph-19-03052-t001:** Descriptive statistics of studied TMD patient cohort and its subgroups (i.e., the study group of TMD patients with a self-reported headache because of problems with teeth, mouth, jaws, or dentures (HATMJD)) and the second subgroup (i.e., the control group of TMD patients without self-reported HATMJD).

Patients’ Characteristics	Mean ± SD or n (%)			*p-*Value
	All TMD pts	Study Group TMD pts w HATMJD	Control Group TMD pts w/o HATMJD	
**No. of patients**	177 (100)	109 (61.6)	68 (38.4)	
**Sociodemographic**				
Female gender	137 (77.4)	92 (84.4)	45 (66.2)	0.005 *
Age	36.3 ± 13.7	35.07 ± 12.2	38.29 ± 15.7	0.153
**Marital status**				0.637
Single/widowed/divorced	69 (39.0)	41 (37.6)	28 (41.2)	
Married/in relationship	108 (61.0)	68 (62.4)	40 (58.8)	
**Education**				0.626
No college	87 (49.1)	52 (47.7)	35 (51.5)	
One year of college or higher	90 (50.9)	57 (52.3)	33 (48.5)	
**Depression (PHQ-9 summary score)**	5.8 ± 5.0	6.6 ± 5.3	4.6 ± 4.1	0.011 *
No depression	86 (48.6)	47 (43.1)	39 (57.4)	
Mild depression	62 (35.0)	41 (37.6)	21 (30.9)	
Moderate depression	18 (10.2)	13 (11.9)	5 (7.4)	
Moderately severe depression	7 (3.9)	4 (3.7)	3 (4.4)	
Severe depression	4 (2.3)	4 (3.7)	/	
**Anxiety (GAD-7 summary score)**	6.0 ± 5.1	6.7 ± 5.2	4.9 ± 4.8	0.020 *
No anxiety	80 (45.2)	43 (39.4)	37 (54.4)	
Mild anxiety	62 (35.0)	42 (38.5)	20 (29.4)	
Moderate anxiety	20 (11.3)	15 (13.8)	5 (7.4)	
Severe anxiety	15 (8.5)	9 (8.3)	6 (8.8)	
**Physical Symptoms (PHQ-15 summary score)**	7.4 ± 4.4	8.4 ± 4.4	6.0 ± 4.1	0.001 *
No somatic symptoms	47 (26.5)	20 (18.3)	27 (39.7)	
Low somatic symptom severity	81 (45.8)	52 (47.7)	29 (42.6)	
Medium somatic symptom severity	37 (20.9)	27 (24.8)	10 (14.7)	
High somatic symptom severity	12 (6.8)	10 (9.2)	2 (2.9)	
**Oral Behaviors (OBC summary score)**	23.9 ± 8.6	26.1 ± 8.0	20.2 ± 8.5	<0.001 *
None	/	/	/	
Low	105 (59.3)	52 (47.7)	53 (77.9)	
High	72 (40.7)	57 (52.3)	15 (22.1)	
**Sleep Quality (PSQI summary score)**	6.0 ± 3.6	6.8 ± 3.7	4.6 ± 2.8	<0.001 *
Good sleep quality	80 (45.2)	38 (34.9)	42 (61.8)	
Poor sleep quality	97 (54.8)	71 (65.1)	26 (38.2)	

TMD: Temporomandibular disorders; PHQ-9: Patient Health Questionnaire-9; GAD-7: General Anxiety Disorder-7; PHQ-15: Patient Health Questionnaire-15; OBC: Oral Behaviors Checklist; PSQI: Pittsburgh Sleep Quality Index. * *p* < 0.05.

**Table 2 ijerph-19-03052-t002:** Comparing the influence of depression, anxiety, functional disorders, oral behaviors, and sleep quality on headache because of problems with teeth, mouth, jaws, or dentures in TMD patients using standardized mean effect reported as Cohen’s d.

Variable	Group	N	Cohen’s d	95% CI for Cohen’s d	
				Lower	Upper
**Sociodemographic**					
Gender	Male TMD pts w HATMJD vs. Female TMD pts w HATMJD	40 vs. 137	0.62	0.26	0.97
Age	<40 y vs. ≥40 y	120 vs. 57	0.07	−0.24	0.39
Marital status	Single/widowed/divorced vs. Married/in relationship	69 vs. 108	<0.00	−0.03	0.03
Education	No college vs. ≥1 years of college	87 vs. 90	0.15	−0.14	0.45
**DC/TMD Axis II diagnoses**					
Depression	No vs. present	86 vs. 91	0.35	0.05	0.64
Physical symptoms	No vs. present	47 vs. 130	0.62	0.28	0.96
Anxiety	No vs. present	80 vs. 97	0.45	0.15	0.75
Parafunctional behaviors	None to low vs. high	105 vs. 72	0.70	0.39	1.01
**Other constructs**					
Sleep quality	Good vs. poor	80 vs. 97	0.64	0.34	0.94

TMD: Temporomandibular disorders; w HATMJD: with a headache because of problems with teeth, mouth, jaws, or dentures; DC/TMD: Diagnostic Criteria for Temporomandibular Disorders.

**Table 3 ijerph-19-03052-t003:** Simple logistic regression analysis predicting the presence of self-reported headache because of problems with teeth, mouth, jaws, or dentures in TMD patients.

Predicting Factor	Odds Ratio	*p*-Value	95% CI for Odds Ratio	
			Lower	Upper
Female gender	2.77	0.006 *	1.34	5.69
Age	0.98	0.130	0.96	1.00
Marital status	0.64	0.637	0.46	1.60
Higher level of education	1.16	0.626	0.63	2.13
Depression	1.10	0.015 *	1.02	1.18
Anxiety	1.08	0.023 *	1.01	1.15
Physical symptoms	1.15	0.001 *	1.06	1.25
Oral behaviors	1.20	<0.001 *	1.09	1.34
Sleep quality	1.24	<0.001 *	1.11	1.39

* *p* < 0.05.

**Table 4 ijerph-19-03052-t004:** Multiple logistic regression analysis, with stepwise forward variable selection method with a headache because of problems with teeth, mouth, jaws, or dentures presence as the outcome.

Step #	Predicting Variable	β Coefficient	*p*-Value	Odds Ratio	95% CI for Odds Ratio	
					Lower	Upper
Step 1	Age	−0.010	0.415	0.990	0.967	1.014
	Female gender	0.932	0.018 *	2.538	1.174	5.486
	OBC	0.086	<0.001 *	1.090	1.043	1.138
	Constant	−1.857	0.018	0.156		
Step 2	Age	−0.015	0.243	0.985	0.961	1.010
	Female gender	0.992	0.014 *	2.696	1.218	5.964
	OBC	0.056	0.026 *	1.058	1.007	1.111
	PSQI	0.156	0.015 *	1.169	1.031	1.325
	Constant	−1.857	0.017	0.148		
Step 3	Age	−0.015	0.243	0.985	0.961	1.010
	Female gender	0.963	0.022 *	2.619	1.149	5.973
	OBC	0.055	0.028 *	1.057	1.006	1.111
	PSQI	0.147	0.045 *	1.158	1.003	1.338
	PHQ-15	0.014	0.804	1.014	0.908	1.132
	Constant	−1.857	0.017	0.147		
Step 4	Age	−0.018	0.170	0.982	0.958	1.008
	Female gender	0.949	0.025 *	2.583	1.129	5.910
	OBC	0.059	0.021 *	1.061	1.009	1.116
	PSQI	0.188	0.021 *	1.207	1.029	1.416
	PHQ-15	0.038	0.522	1.039	0.924	1.168
	PHQ-9	−0.073	0.204	0.930	0.831	1.040
	Constant	−1.892	0.019	0.151		
Step 5	Age	−0.018	0.159	0.982	0.957	1.007
	Female gender	0.969	0.023 *	2.636	1.145	6.070
	OBC	0.061	0.019 *	1.063	1.010	1.118
	PSQI	0.188	0.021 *	1.207	1.029	1.415
	PHQ-15	0.038	0.528	1.039	0.923	1.168
	PHQ-9	−0.048	0.549	0.953	0.815	1.115
	GAD-7	−0.028	0.659	0.972	0.857	1.102
	Constant	−1.892	0.019	0.151		

PHQ-9: Patient Health Questionnaire-9; GAD-7: General Anxiety Disorder-7; PHQ-15: Patient Health Questionnaire-15; OBC: Oral Behaviors Checklist; PSQI: Pittsburgh Sleep Quality Index. * *p* < 0.05. The goodness of fit: χ^2^ < 0.05.

## Data Availability

The datasets generated during the current study are available from the corresponding author on reasonable request.

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
