# Peer review of "Headache Because of Problems with Teeth, Mouth, Jaws, or Dentures in Chronic Temporomandibular Disorder Patients: A Case–Control Study"

_ijerph, 2022, doi:10.3390/ijerph19053052_

Round 1
Reviewer 1 Report
Dear editors,
Thanks for giving me the opportunity to review the manuscript of Ostrc et al. about TMJ pain and its related consequences.
To begin with, the used language needs comprehensive improvements. There is a problem with using “the English articles” (the, a, an) in the whole manuscript. The paper was not structured well and it is very difficult to follow its ideas.
Both the title and the abstract are very wordy and non-representable and I had to read them several times to understand the aims and the major findings of the study.
The followings are some comments that may help the authors improve the manuscript:
- line 20: I would prefer if you start this statement as “ exclusion criteria included ……” it would be more clear
- line 117: I don’t understand what does “consecutive” in this context mean as it seems irrelevant.
- line 120: “a board-certified…” instead of “the board-certified”
- line 139: “44 Patients are asked how often…” apparently the purpose of this statement is to justify or explain PHQ-9. However, this should not be mentioned in the methodology, instead, this can be added to the discussion part
- line 138 to 181: same as the above point, the authors provided very detailed explanations to justify using specific scales to measure depression, anxiety, parafunction …. . however, I believe that the methodology part should include only technical details about the used scales and how the variables were measured, not providing background about the significance of these scales and their efficacy to measure certain subjective variables. In my opinion, these details should be discussed in the discussion part rather than the method one.
- line 182 2.3 methods: what does “methods” as a subtitle here mean? as you already talking about the materials and methods. this could be named data collection for example
- line 226. the mean age in this line is for all included subjects or only for women? please clarify this point.
- The result section is very wordy and does not highlight the major outcomes clearly.
Reviewer 2 Report
The present clinical study, investigating self-reported HATMJD in chronic TMD subject, comparing their results with those of TMD patients without such headaches, and investigating the associations of HATMJD with depression, anxiety, physical symptoms, oral behaviors and sleep quality, is very interesting and well described.
The manuscript is well organized and written, although a minor revision for English language is required.
Results are clearly presented. Discussion section is well structured. Methods needs to be implemented.
Reviewer’s suggestions are given below:
- Please, clarify if in this case-control study, cases and controls were matched for age and gender and if the two groups were composed of the same number of subjects;
- Was the sample size computed? Is the number of participants adequate for the analysis? If not, please, enroll more subjects;
- Please, remove “Study instrument” (line 125).
Reviewer 3 Report
Congratulations to the authors. The manuscript is well-written and conceptual and methodologically is very appropriate. I recommend the publication of this manuscript
Round 2
Reviewer 1 Report
Dear authors, thanks for considering my previously listed points. I am happy with the current version without further modifications. Congratulations
Reviewer 2 Report
I would suggest to accept the manuscript in the current form